# A Joint Automatic Modulation Classification Scheme in Spatial Cognitive Communication

**DOI:** 10.3390/s22176500

**Published:** 2022-08-29

**Authors:** Mengtao Wang, Youchen Fan, Shengliang Fang, Tianshu Cui, Donghang Cheng

**Affiliations:** 1School of Space Information, Space Engineering University, Beijing 101416, China; 2School of National Space Science Center, Chinese Academy of Sciences, Beijing 101416, China

**Keywords:** automatic modulation classification, convolutional neural networks, spatial cognitive communication, deep learning, expert feature methods

## Abstract

Automatic modulation discrimination (AMC) is one of the critical technologies in spatial cognitive communication systems. Building a high-performance AMC model in intelligent receivers can help to realize adaptive signal synchronization and demodulation. However, tackling the intra-class diversity problem is challenging to AMC based on deep learning (DL), as 16QAM and 64QAM are not easily distinguished by DL networks. In order to overcome the problem, this paper proposes a joint AMC model that combines DL and expert features. In this model, the former builds a neural network that can extract the time series and phase features of in-phase and quadrature component (IQ) samples, which improves the feature extraction capability of the network in similar models; the latter achieves accurate classification of QAM signals by constructing effective feature parameters. Experimental results demonstrate that our proposed joint AMC model performs better than the benchmark networks. The classification accuracy is increased by 11.5% at a 10 dB signal-to-noise ratio (SNR). At the same time, it also improves the discrimination of QAM signals.

## 1. Introduction

As NASA enters a new era of space exploration, communication links shift from point-to-point communications to network topologies. There are more diverse types of wireless links in space communications, such as planetary earth to earth, planetary earth to spacecraft, and space to earth, etc. [1]. To improve situational awareness, we seek to develop an ACM algorithm based on DL that is capable of identifying common signals in satellite communications and thus can efficiently identify users and distinguish interference sources.

Traditional AMC methods are mainly divided into two categories: the likelihood-based (LB) AMC [2,3] and the feature-based (FB) AMC [4,5,6]. The LB-AMC approach is based on the Bayesian theory in order to obtain the best estimate of modulation by minimizing the probability of misclassification, but it has the disadvantages of high computational complexity and narrow applicability. The purpose of FB-AMC is to find features that can distinguish different modulated signals, such as wavelet domain features [4], cyclic spectrum [5], and high-order statistics [6]. Furthermore, the performance of the FB classifier is significantly influenced by the quality of the features.

DL is a data-driven artificial intelligence approach that uses multilayer neural networks to extract data features automatically. O’Shea et al. [7] first proposed using a convolutional neural network (CNN) to process the IQ signals directly, and the average recognition rate was 75% at 10 dB in the RadioML 2016.10a dataset which includes 11 modulation classes. However, this CNN network only has two layers, so its classification performance is limited. Subsequently, O’Shea et al. [8] proposed an improved ResNet network to improve recognition performance. In addition, multiple deep CNNs were applied to boost the performance of AMC in [9,10,11]. However, these CNNs mostly used convolutional kernels with 3 × 1 dimensions, which cannot capture the long-term temporal features of IQ signals. Meanwhile, West [12] et al. first proposed a CLDNN network combining a CNN network and a long short-term memory (LSTM) network, which can extract long-term temporal features. This network got an average recognition accuracy of 85% at 0 dB in the RadioML 2016.10a dataset.

More and more neural networks are being used to improve AMC’s performance. However, these networks are not very good at identifying intra-class diversity signals. Yu Wang et al. [13] proposed a data-driven fusion model which combines two CNN networks, one trained on the IQ signal dataset, and the other trained on the constellation map dataset. Inspired by face recognition, Hao Zhang et al. [14] proposed a two-stage training network that improves the model’s ability to capture small intra-class scattering. The central loss function supervises the first stage, and the cross-entropy loss function supervises the second stage. Kumar Yashashwi et al. [15] used an attention model to synchronize and normalize signals, which improves the model’s recognition of intra-class diversity signals. However, these works all face the problem of poor generalization ability. If we substitute another dataset, these methods may not be applicable.

Summarizing the previous work, we can find that the improvement of neural networks in AMC is achieved by improving the ability to extract signal timing features. However, IQ signals contain not only timing features but also phase features. Therefore, when building a neural network, we consider extracting both the timing and phase features of the signal. In addition, the neural network is weak in extracting intra-class features. Thus, cascading a network trained on different datasets [13] or cascading a network supervised by other loss functions [14] still results in limited generalizability. Therefore, we choose to group 16QAM and 64QAM signals with similar intra-class features into the same class and identify them through use of the expert feature method so as to solve the problem in disguise.

## 2. System Model

### 2.1. AMC-Driven Intelligent Receiver Architecure

Figure 1 shows a satellite intelligent receiver system based on a zero-IF architecture. The AMC-driven intelligent receiver can identify the modulation type of the original signal without any prior information. Moreover, it can help subsequent modules, such as symbol synchronization, channel equalization, and signal demodulation [16]. The workflow of this intelligent receiver is as follows: the RF signal first passes through the mid-pass filter BPF and low-noise amplifier LNA for frequency selection and amplification. Then, the signal is sent to the mixer and the local oscillator frequency for mixing to generate the in-phase component I and the quadrature component Q. Next, the I and Q signals are amplified, filtered, sampled and extracted to create the digital IQ baseband signal. Finally, the acquired IQ baseband signal is input into the AMC model to complete the identification of signal modulation type.

### 2.2. The Joint AMC Model

This paper proposes a joint automatic identification model that combines the IQCLNet network and expert feature methods. As shown in Figure 2, the model is used to identify 11 modulated signals widely used in modern communication systems. When the receiver acquires the unknown signals, the first stage will be made by IQCLNet to identify them. In addition, QAM16 and QAM64 are considered the same class and named QAMS in this stage. Then, the second stage uses the expert feature method to construct parametric features in order to identify QAMS.

## 3. Method of Proposing Models

### 3.1. IQCLNet Network

In electromagnetic signal recognition, most DL network structures are borrowed from the network design in image identification. In image processing, the input pixel data format is M × N, which has an isotropic nature in the spatial relationship. Thus, the shape of the convolution kernel is generally square to perform the symmetric operation between two dimensions. However, in signal processing, the input IQ data format is N×. N is the time sampling point, reflecting the time series characteristics. Two corresponds to I and Q, reflecting the phase characteristics [17]. IQ data do not have the same nature between the two dimensions, so they cannot be operated symmetrically as image processing.

At present, the processing of IQ signals by DL networks mostly uses one-dimensional convolution kernels to extract the time-series features of the signals [7,18,19,20] while ignoring the phase characteristics, so we must design a network which can extract the different features in two dimensions of the IQ signals. In this paper, we create the network structure as shown in Figure 3. Within each channel, the first convolutional layer uses a 1 × 2 convolutional kernel to extract the phase features of the signal, and the output data dimension is changed from N × 2 to N × 1; then, a 3 × 1 convolutional kernel is used to extract the short-time sequence features of the signal, followed by a cascaded layer of LSTM to extract the long-time sequence features of the signal [21]. Finally, the fully connected layer maps the output data to a more discrete space for classification.

In the classification stage, adoption of the adaptive average pooling layer occurs first, and the mean value of each channel eigenvalue is used as a new eigenvalue that not only can reduce the parameters of the fully connected layer, but also improve the generalization performance of the network. Next, using only one fully connected layer for classification in order to reduce the number of parameters and the amount of computation, the specific implementation is: first set the output value of the adaptive average pooling to be consistent with the number of channels C, and then form a fully connected layer with the input C and the output L with the number of categories L. This operation makes the network compatible with different input lengths’ IQ signal. Between the output of the convolutional layer and the activation function, a Batch Normalization (BN) operation is added to increase the robustness and convergence speed of the model; the feature extraction layer uses ReLU as the activation function and Softmax as the output function of the classification layer.

The specific parameters of the IQCLNet network are shown in the following Table 1:

### 3.2. Expert Feature Method

The amplitude-phase modulated signal model of the QAM signal at the receiver side is expressed as:(1)$$r(t)=e^{j2\pi f_{c}t}e^{j\theta_{c}}\sum_{i=1}^{N}\alpha_{i}g\left(t-iT_{0}-\varepsilon T_{0}\right)+\omega (t)$$
(2)$$\alpha_{i}=\sqrt{S_{i}}e^{j\varphi_{i}}$$
(3)$$S_{i}=\sqrt{\alpha_{i,I}^{2}+\alpha_{i,Q}^{2}}$$
(4)$$\varphi_{i}=\arctan\frac{\alpha_{i,Q}}{\alpha_{i,I}}$$
where *r*(*t*) is the received signal; *g*(*t*) is the shock response of the shaping filter; *T*_0_ is the codeword period; *f_c_* is the carrier frequency; *θ_c_* is the carrier phase; *ε* is the timing offset; *N* is the number of observation symbols; *α_i_* is the zero-mean smooth complex random sequence, i.e., the transmit codeword sequence; $\sqrt{s}$ and *φ_i_* are the amplitude and phase of *α_i_*, respectively; and *ω*(*t*) is a stationary additive Gaussian noise with a zero mean and a one-sided power spectral density *N*_0_ [22].

QAM signal modulation information is not only reflected in the phase variation but also in the amplitude variation. However, QAM signals have many types of phase change, which should not be suitable for intra-class identification. Under ideal conditions, the number of 16QAM, 64QAM, and 256QAM signal amplitude takes 3, 9, and 32, which have significant differences. The authors in [23] mention that the zero-center normalized instantaneous amplitude tightness characteristic parameter (*μ*_42_) reflects the denseness of the instantaneous amplitude distribution. Therefore, we can use *μ*_42_ to distinguish each order of QAM signals. *μ*_42_ is defined in Equation (8), where *a*_c*n*_ is the zero-center normalized instantaneous amplitude.
(5)$$m_{a}=\frac{1}{N_{S}}\sum_{i=1}^{N}a(i)$$
(6)$$a_{n}(i)=\frac{a(i)}{m_{a}}$$
(7)$$a_{cn}\left(i\right)=a_{n}\left(i\right)-1$$
(8)$$\mu_{42}^{\alpha }=\frac{E\left[\alpha_{\mathrm{cn}}^{4}(i)\right]}{{\left\{E\left[\alpha_{\mathrm{cn}}^{2}(i)\right]\right\}}^{2}}$$

## 4. Experimental Results

### 4.1. IQCLNet Network Experiments

#### 4.1.1. Dataset

In this paper, we use a popular open-source dataset Radio ML2016.10a [24]. This dataset has 11 classes of modulated signals with SNR ranging from −20 dB to 18 dB, and the length of a single sample is 128. The details are shown in Table 2. All experiments are conducted on this dataset.

#### 4.1.2. The Superiority of IQCLNet Network

Figure 4 demonstrates three feature extraction methods for IQ signals processed by convolutional neural networks. Among them, (a) is the method adopted in this paper, (b) is the method adopted in [18,19], and (c) is the method adopted in [7,20]. The experimental results of the three different convolution methods on the dataset are shown in Figure 5. The average recognition rates of (a), (b), and (c) are 62.8%, 58.2%, and 59.4%, respectively. (a) As the structure proposed in this paper, phase feature extraction is performed on the signal first, and then timing feature extraction is performed, so that it has the highest recognition rate. (b) first convolves with a 1D filter and then flattens the data, but this method can only extract time-domain features and cannot use phase features; (c) is the same as the first step in (b). After extracting temporal features, (c) performs dimensionality reduction in the IQ direction using MaxPooling. However, this method loses the amplitude and phase information of the signal. Furthermore, the convolution method (a) used in this paper shows better recognition results in both low SNR and high SNR conditions. Thus, the superiority of IQCLNet in this paper is verified.

#### 4.1.3. The Effectiveness of IQCLNet Network

The joint AMC model we designed requires that the IQCLNet network can identify the QAMS effectively, so that the subsequent expert feature method can identify 16QAM and 64QAM accurately. Therefore, we provide the confusion matrices of IQCLNet under four different SNR conditions. The confusion matrix is a method of accuracy evaluation. The column represents the predicted category, the row represents the real category, and the darker the color of the square where the row and column intersect, the higher the accuracy is. As shown in Figure 6, the network cannot identify any signal under the extremely low SNR of −20 dB. When the SNR is −12 dB, the QAMS signal can already be distinguished from other signals. With the SNR improvement, the QAMS signal recognition can reach 100%. These experimental results demonstrate the effectiveness of the IQCLNet network.

### 4.2. Expert Feature Method Experiments

After solving the experimental verification of the IQCLNet network, we need to construct a classifier to distinguish the QAM signal. First, we should calculate the *μ*_42_ of 16QAM and 64QAM signals at different SNRs. The two signals’ feature parameter curves are shown in Figure 7a. We can distinguish the two signals clearly through the *μ*_42_ feature parameter. Thus, taking the average of the *μ*_42_ of the two signals as the threshold, according to the size relationship between *μ*_42_ and the threshold line, we can distinguish 16QAM and 64QAM. Moreover, the two curves intersect at 0 dB, which may make the two signals difficult to distinguish around this SNR.

Now we have constructed a classifier that can distinguish QAM signals. Next, we use it to identify the QAMS signal output by IQCLNet. At the same time, we use IQCLNet to identify the QAMS signal directly for comparison. The experimental results are shown in Figure 7b.

The IQCLNet method’s recognition accuracy steadily improves with the increase of SNR, and it tends to be stable under high SNR conditions. Moreover, the overall average recognition rate is 60.4%. Compared with the IQCLNet network, the recognition effect of the expert feature method is significantly improved under low SNR conditions. However, since the characteristic curves of 16QAM and 64QAM intersect around 0 dB, the recognition rate will drop. Moreover, the overall average recognition rate is 77.9%, an increase of 17.5% compared to exclusive use of the IQCLNet method. This proves that the expert feature method has obvious advantages in identifying QAM signals.

### 4.3. The Joint AMC Model Results

We can easily derive the total recognition rate of the joint AMC model after getting the recognition rate of QAMs. We select three models in Table 3 as comparison networks. CNN2 [7] is the first classical structure that uses a convolutional neural network to recognize modulation; CLDNN [12] is a classical structure in speech recognition tasks that has been successfully transplanted into the field of electromagnetic signal recognition. CNN_LSTM [25] is a well-designed network structure based on CLDNN which uses fewer parameters and obtains higher recognition accuracy. All three networks have been validated on the RML2016.10a dataset.

The classification accuracies of all models are shown in Figure 8, taking 0 dB as the dividing line between high and low SNR. Under low SNR conditions, the average recognition rates of the three baseline networks are 30.8%, 29.1%, and 31.1%, respectively. The IQCLNet network and Joint AMC model are 34.1% and 41.7%. Under high SNR conditions, the average recognition rates of the three baseline networks are 74.1%, 79.6%, and 83.4%, respectively. The IQCLNet network and Joint AMC model are 89.2% and 90.9%. Experimental results show that, compared with the three baseline networks that only extract the timing features of the IQ signal, the IQCLNet, with its additional phase feature extraction, is more effective. Moreover, since the recognition rate of 16QAM and 64QAM is improved. The joint AMC model that adds an expert feature method after the IQCLNet network is further improved, especially at low SNR.

In addition, we provide the confusion matrixes of IQCLnet and the Joint AMC model at 10dB SNR in Figure 9a,b. It can be seen that the joint AMC model improves the recognition ability of 16QAM and 64QAM.

Due to the limitations of volume, mass, and power consumption, and the influence of environmental factors such as space radiation and extreme temperature, the computing power and storage space of space-borne computers are very different from those of ground-based computers. Although deep neural networks have the advantages of strong feature extraction ability and high recognition accuracy, they also have the disadvantages of many network parameters and a large amount of calculation. Therefore, we compare the number of parameters and training time of all networks, and the experimental results are shown in Table 4. Compared with other networks, our proposed IQCLNet network has fewer parameters and higher computational efficiency, which is more conducive to deployment to satellite in-orbit applications.

## 5. Conclusions

In this paper, we propose an innovative joint AMC model to identify different modulated signals. The model is based on the high performance of the forward deep learning network IQCLNet, which can separate the QAMs accurately. Then, expert feature methods are used to construct feature parameters in order to identify 16QAM and 64QAM. It is concluded that the joint AMC model exhibits better recognition performance and intra-class diversity discrimination ability than the baseline network. In future research, we can consider communication as an end-to-end reconstruction optimization task, and use autoencoders to learn channel models, encoding and decoding implementations without prior knowledge.

## Figures and Tables

**Figure 1 sensors-22-06500-f001:**
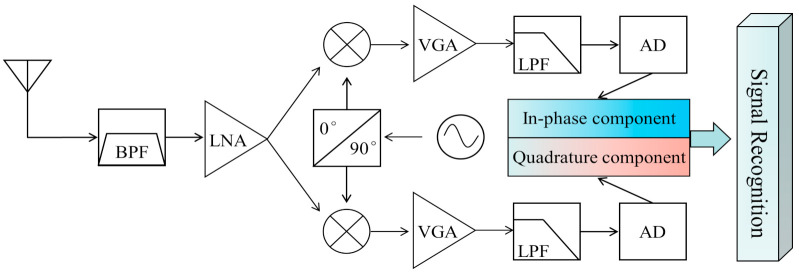
Quadrature sampling zero-IF intelligent receiver.

**Figure 2 sensors-22-06500-f002:**
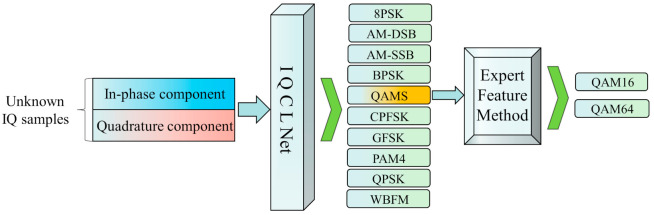
The Joint AMC model.

**Figure 3 sensors-22-06500-f003:**
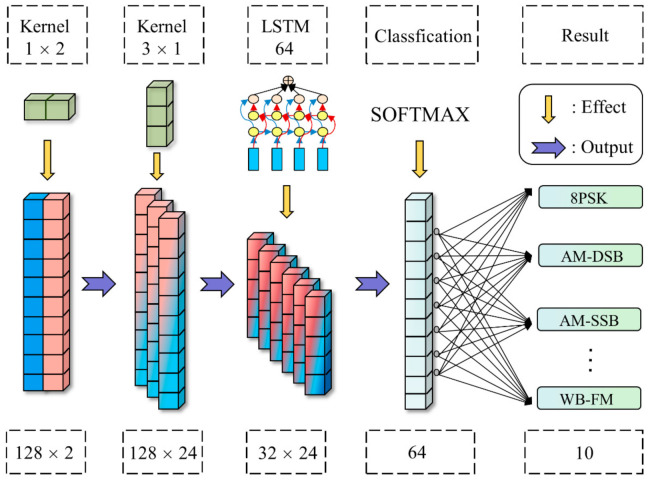
IQCLNet network model.

**Figure 4 sensors-22-06500-f004:**
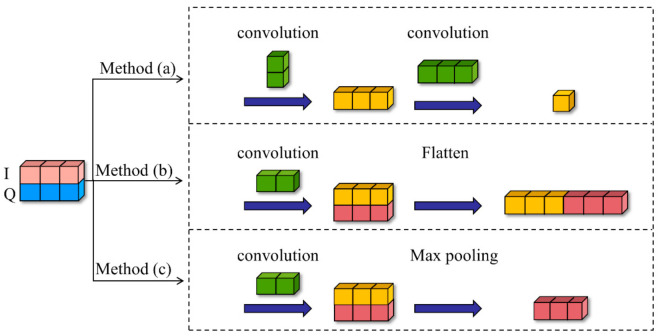
Three convolution methods.

**Figure 5 sensors-22-06500-f005:**
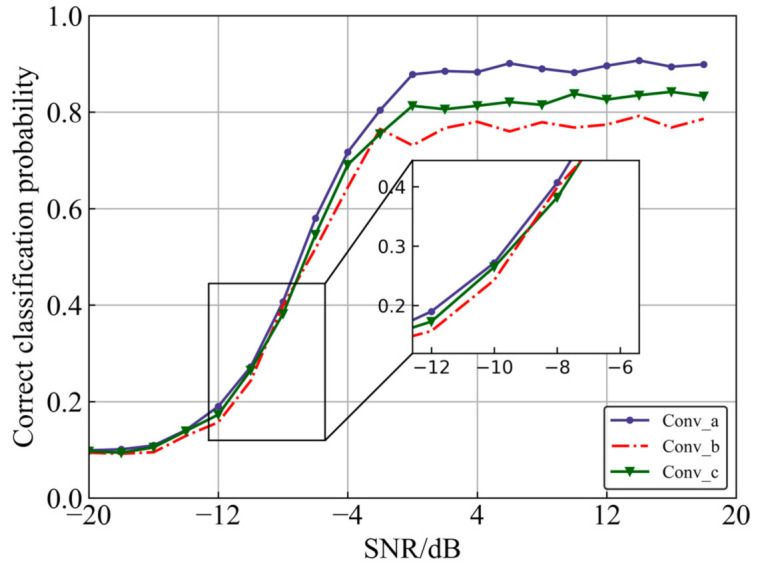
Results of different convolution methods.

**Figure 6 sensors-22-06500-f006:**
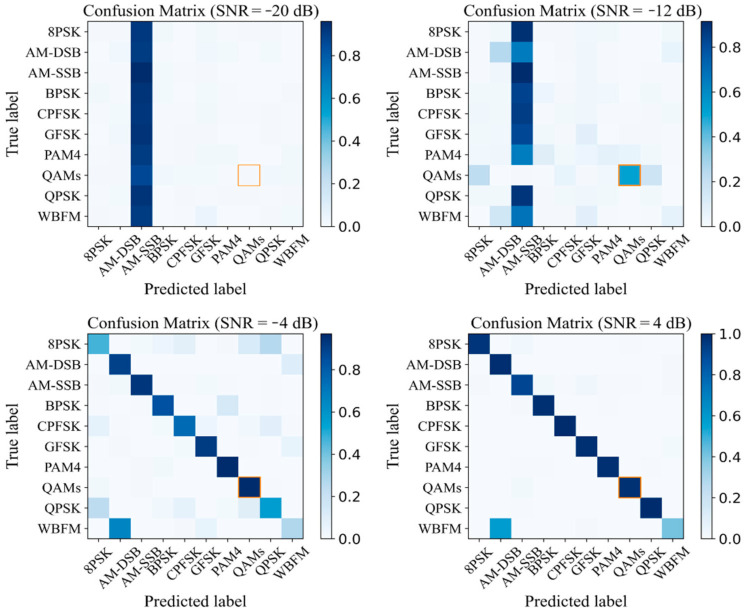
Confusion matrixes of IQCLNet at different SNR ratios.

**Figure 7 sensors-22-06500-f007:**
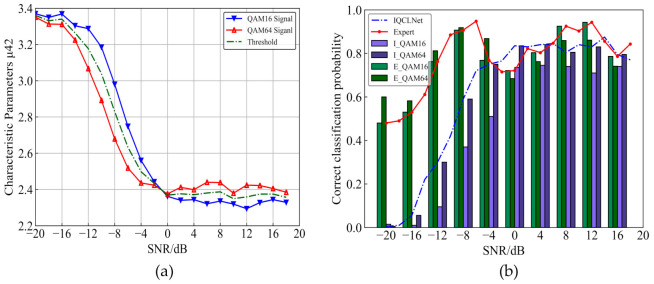
Experimental results of the expert feature method. (**a**) Characteristic parameter *μ*_42_; (**b**) results of IQCLNet and expert feature methods in identifying QAM, respectively.

**Figure 8 sensors-22-06500-f008:**
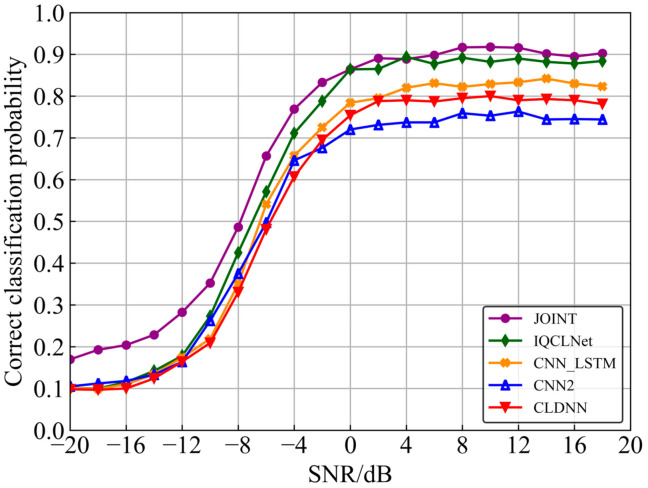
Classification accuracy of different networks.

**Figure 9 sensors-22-06500-f009:**
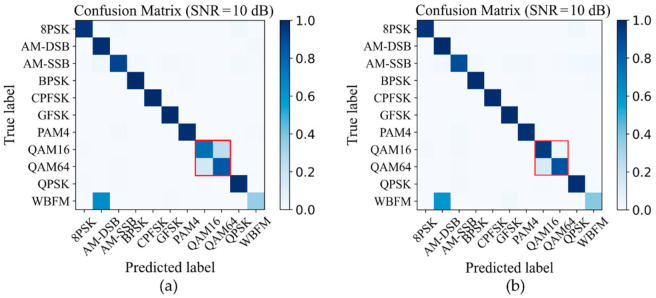
Confusion matrixes of the two methods: (**a**) IQCLNet network; (**b**) the joint AMC model.

**Table 1 sensors-22-06500-t001:** The structure of the IQCLNet network.

Layer Type	Input Size	Output Size	Details
Input	1 × 2 × 128	-	-
Convolution	1 × 2 × 128	24 × 1 × 128	Kernel Size: 1 × 2 Activation: Relu
Lambda	24 × 1 × 128	24 × 128	Squeeze (x, axis = 2)
Convolution	24 × 128	24 × 128	Kernel Size: 3 × 1 Activation: Relu
MaxPool	24 × 128	24 × 64	Pool Size: 2 Strides = 2
Convolution	24 × 64	24 64	Kernel Size: 3 × 1 Activation: Relu
MaxPool	24 × 64	24 × 32	Pool Size: 2 Strides = 2
LSTM	24 × 32	64	Units: 64
Dense + Softmax	64	10	One-Hot Output

**Table 2 sensors-22-06500-t002:** RML2016.10a dataset.

Dataset	RadioML2016.10a
Modilations	8 Digital Modulations: BPSK, QPSK, 8PSK, 16QAM,64QAM, BFSK, CPFSK, and PAM4 3 Analog Modulations: WBFM, AM-SSB, and AM-DSB
Length per sample	128
Signal format	In-phase and quadrature (IQ)
Sampling frequency	1 MHz
SNR Range	[−20 dB, −18 dB, …, 18 dB]
Total number of samples	220,000 vectors

**Table 3 sensors-22-06500-t003:** Structural parameters of different networks.

MODEL	CNN2	CLDNN	CNN_LSTM
Convolution Layers	2	3	2
Kernel Size	1 × 2, 2 × 3	1 × 8	1 × 3, 2 × 3
Convolution Channels	256, 80	50 × 3	128, 32
LSTM Layers	0	1	1
LSTM Units	0	50 × 1	128 × 1

**Table 4 sensors-22-06500-t004:** Parameters and computer time of different networks.

MODEL	Network Parameters	Training Time (s)
CNN2	5,145,936	1037
CLDNN	69,253	2430
CNN_LSTM	113,082	1523
IQCLNet	30,471	301

## Data Availability

Not applicable.

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
