# Peer review of "A Joint Automatic Modulation Classification Scheme in Spatial Cognitive Communication"

_sensors, 2022, doi:10.3390/s22176500_

Round 1

Reviewer 1 Report

The article is interesting, well presented, contains enough theoretical background and explaining illustrations. Authors should work on the grammar and English language through the text. After improvement of language mistakes and some sentences structures, the article can be recommended for publication.

Reviewer 2 Report

1. The abstract does not highlight the novelty of the paper. It is written as of a report .

2. The first statement in the introduction is out of date,  refer to latest references.

3. Introduction, last paragraph ' Summarizing the previous work...' , include the references to this statement. What is the differences between this work and the previous, This need to be highlighted here.

4.  System Model 2.1 and 2.2 there is not much information, this is more towards textbook writing. highlight on the importance and how does this implies towards your work.

5. change 'Methond' to 'Method'

6. How did equations 1 to 4 came about. Did the authors derive or referred? 

7. The method of the proposed model is very brief. Need to show in detail the algorithm developed. 

8. open-source dataset Radio ML2016.10a is very old, why new data has not been used?

9.All the figures explanation are simple, include constructive explanation.

10. There should be intense comparison of the developed method with existing

Reviewer 3 Report

An interesting DL-based modulation classification scheme has been proposed in the paper. The test results in Section 4 had demonstrate quite well both the application and accuracy of the proposed technique. The proposed method appears technically sound and would be of significant interest to those working in the same field of research. 

In Section 1 of the introduction, the performance of the classifiers developed by other researchers had been mentioned but only the overall average result was stated. It is somewhat difficult to compare this with the results in Section 4. It may be better to compare in detail the performance of those other classifiers with the one proposed in this paper together in Section 4. Furthermore, if the results from the other classifiers were using a different dataset, then I recommend using the same dataset in the comparison.

In addition, I recommend clarifying to the reader the motivation behind the need for such classifiers. There were only a couple of sentences mentioned in Section 1 on this matter, but they don't appear to strongly highlight the importance or the need for such solutions. 

Round 2

Reviewer 2 Report

All corrections have been done